# PLANCKIAN JITTER: ENHANCING THE COLOR QUALITY OF SELF-SUPERVISED VISUAL REPRESENTATIONS

## ABSTRACT

Several recent works on self-supervised learning are trained by mapping different augmentations of the same image to the same feature representation. The set of used data augmentations is of crucial importance for the quality of the learned feature representation. We analyze how the traditionally used color jitter negatively impacts the quality of the color features in the learned feature representation. To address this problem, we replace this module with physics-based color augmentation, called Planckian jitter, which creates realistic variations in chromaticity, producing a model robust to llumination changes that can be commonly observed in real life, while maintaining the ability to discriminate the image content based on color information. We further improve the performance by introducing a latent space combination of color-sensitive and non-color-sensitive features. These are found to be complementary and the combination leads to large absolute performance gains over the default data augmentation on color classification tasks, including on Flowers-102 (+15%), Cub200 (+11%), VegFru (+15%), and T1K+ (+12%). Finally, we present a color sensitivity analysis to document the impact of different training methods on the model neurons and we show that the performance of the learned features is robust with respect to illuminant variations.

## 1 INTRODUCTION

Self-supervised learning enables the learning of visual representation without the need for any labeled data (Doersch et al., 2015; Dosovitskiy et al., 2014). Several recent works learn representations that are invariant with respect to a set of data augmentations, and have obtained spectacular results (Grill et al., 2020; Chen & He, 2021; Caron et al., 2020), significantly narrowing the gap with supervised learned representations. These works vary in their architecture, learning objective, and optimization strategy, however, they are similar in applying a common set of data augmentations to generate the various image views. The algorithms, while learning to map these different views to the same latent representation, learn complex semantic representation for visual data. The set of transformations (data augmentation) that are considered induce a set of invariances that characterizes the learned visual representation.

Before deep learning revolutionized the way visual representations were computed, separate features were hand-designed to represent its various properties, leading to research on shape (Lowe, 2004), texture (Manjunath & Ma, 1996), and color features (Finlayson & Schaefer, 2001; Geusebroek et al., 2001). Color features were typically designed to be invariant with respect to a set of scene accidental events, such as shadows, shading, illuminant and viewpoint changes. With the rise of deep learning, feature representations that exploit simultaneously color, shape and texture are learned implicitly and the invariances are a byproduct of the end-to-end training (Krizhevsky et al., 2009). As discussed above, the current set of self-supervised learning methods explicitly define a set of invariances (related to the applied data augmentations) that are to be learned. In this work, we focus on the current de-facto choice for color augmentations. We argue that they seriously cripple the color quality of these representations and we propose an alternative color augmentation.

Figure 1 (left) illustrates the currently applied color transformation for a sample image, depicted in the middle of the left-most grid. It is clear that the applied color transformation significantly alters the colors of the original image, both in terms of hue and saturation. One of the justifications in literature for such strong color augmentations is that without large color changes, mapping images to the same latent representation can be purely done based on color and no complex shape features are learned, therefore the best results when using only two transformations are obtained when applying

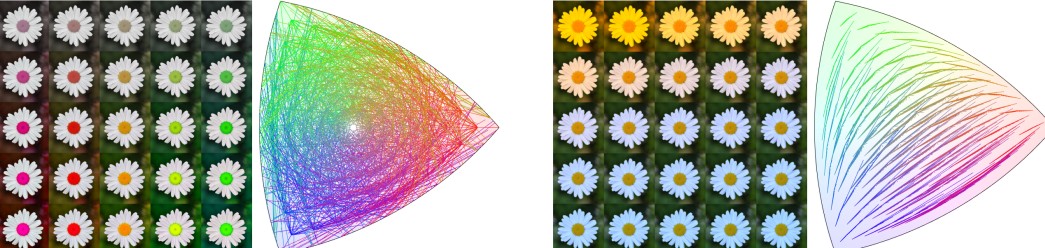

Figure 1: Default color jitter (left) and Planckian jitter (right). Augmentations based on default color jitter lead to unrealistic images, while Planckian augmentations lead to a set of realistic augmentations. We also provide the ARC chromaticity diagrams. Here we sample a number of initial RGB values, and map them into the range of possible outputs given by the respective data augmentation technique. These diagrams show that Planckian jitter transforms colors along chromaticity lines that occur in nature when changing the illuminant, whereas the default color jitter transfers colors throughout the whole chromaticity plane. In both cases, only chromaticity effects have been shown, disregarding brightness and contrast variations, to simplify the visualization.

image cropping with a color augmentation (Chen et al., 2020a). However, looking at the reported example it is evident that a representation that maps these images to the same latent representation cannot rely on the object color, and as a results the quality of the color representation learned with such algorithms is expected to be inferior.

Therefore, in this paper, we propose another set of color augmentations, shown on the right side of Figure 1. In addition to introducing more natural variations in the image chromaticity, the proposed color augmentation also affects neutral regions such as the petals shown in the figure, which are left unvaried with the original color transformations. We draw on existing color imaging literature, that aimed to design features that were invariant with respect to illuminant changes that were commonly encountered in real-world scenes (Finlayson & Schaefer, 2001). Our augmentation, called *Planckian jitter*, applies a physically realistic illuminant variation to the images. We consider the illuminants that are described by Planck's Law for black body radiation and that are known to be similar to illuminants encountered in real-life (Tominaga et al., 1999). In the experimental section, we show that self-supervised features learned with Planckian jitter yield superior features, leading to gains of over 5% on several downstream color classification tasks. However, since our color augmentation is less extreme, the learned shape features are of lower quality than with the original color jitter. A simple combination of both feature representations leads to huge performance gains with respect to default color jitter of between 10-15% on several color downstream tasks. In addition, we show that our augmentation method can be applied to several state-of-the-art self-supervised learning methods. Finally, we analyze the color sensitivity of the learned color representations.

## 2 BACKGROUND AND RELATED WORKS

### 2.1 SELF-SUPERVISED LEARNING AND CONTRASTIVE LEARNING

Recent improvements in self-supervision learn a semantically rich feature representation without the need of any labels. In SimCLR (Chen et al., 2020a) similar samples are created by augmenting an input image, while dissimilar are chosen by random. To make contrastive training more efficient MoCo method (He et al., 2020) and its improved version (Chen et al., 2020b) use a memory bank for learned embeddings which helps for efficient sampling. This memory is kept in sync with the rest of the network during training time by using a momentum encoder. Several methods do not have any explicit contrastive pairs. BYOL (Grill et al., 2020) propose an asymmetric network by introducing an additional MLP predictor between the outputs of the two branches. One of the branches is kept "offline" - updated by a momentum encoder. SimSiam (Chen & He, 2021) goes even further and presents a simplified solution without a momentum encoder. Moreover, it obtains similar high-quality results and does not require a large mini-batch size.

We will use the SimSiam method to verify our proposed color augmentation (we also apply our approach to SimCLR (Chen et al., 2020a) and Barlow Twins (Zbontar et al., 2021) in the experiments). The main part is a CNN-based encoder, learned end-to-end in an asymmetric siamese architecture,

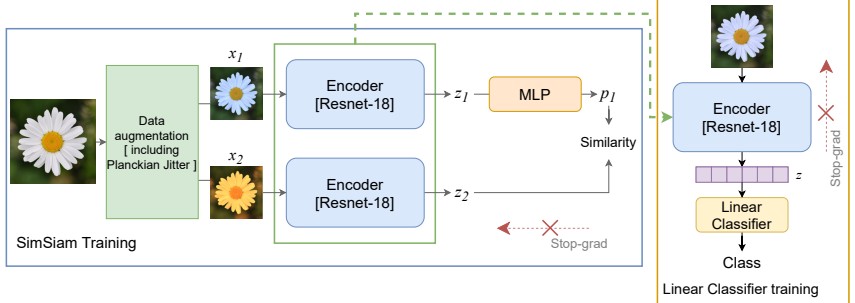

Figure 2: SimSiam training procedure exploiting Planckian-based data augmentation (left), and fine-tuning the linear classifier using the trained encoder (right).

where one branch has an additional predictor (Multi-Layer Perceptron, or MLP) whose output aims to be as close as possible to the other branch (see Figure 2). The second branch is not updated during backward propagation. A negative-cosine loss function is used, defined as:

$$\mathcal{L} = \mathcal{D}(p_1, z_2)/2 + \mathcal{D}(p_2, z_1)/2 \tag{1}$$

$$\mathcal{D}(p_A, z_B) = -\frac{p_A}{\|p_A\|_2} \cdot \frac{z_B}{\|z_B\|_2}, \tag{2}$$

where $z_1$, $z_2$ are encoded values respectively for two different augmented versions $x_1$ and $x_2$ of the same image $x$. Note that in Eq. 1 they are alternated between the two branches, but it is always only the first branch that uses a Multilayer Perceptron, producing either $p_1$ or $p_2$. Additionally, no contrastive term is present: only the similarity is enforced during learning.

## 2.2 DATA AUGMENTATION

Data augmentation plays an important part in the learning process of self-supervised methods. In the works by Chen et al. (2020a) and Zbontar et al. (2021), authors discussed the importance of the different data augmentations. A set of well-defined transformations was proposed within the SimCLR method. This set is commonly used in later works in the self-supervision field. The augmentations include: rotation, cut, flip, color jitter, blur and gray scale. These operations are randomly applied to an image to generate different views ($x_1$, $x_2$). Applied to the same image contrastive-like self-supervised methods learn representation invariant for such distortions. The multiple view creation is a task-related procedure (Tian et al., 2020). However, keeping in mind usefulness of learned representations for downstream tasks, color jittering is one of the most important ones (Chen et al., 2020a; Zbontar et al., 2021), operating on hue, saturation, brightness and contrast. However, color jitter is expected to induce a certain level of color-invariance (invariance to hue, saturation, brightnesss and contrast) which are consequently transferred to the downstream task as well. As a consequence, we expect these learned features to underperform on downstream tasks for which color is crucial.

Color imaging literature has long researched color features that were invariant with respect to scene accidental events, such as shading, shadows, and illuminant changes (Geusebroek et al., 2001; Finlayson & Schaefer, 2001). Having features with invariance with respect to these events was found to be beneficial for object recognition. Having invariance with respect to hue and saturation changes (which are induced by the currently used color jitter operation) is detrimental for object recognition, especially for those objects were these characteristics are fundamental. Therefore, in this work, we aim to revisit early theory on illuminant invariance (Finlayson & Schaefer, 2001) to design an improved color augmentation function that induces invariances common in the real-world and that does not deteriorate the color quality of the learned features.

## 3 PLANCKIAN JITTER

The range of image transformations introduced by traditional color jitter creates a variability in training data that indiscriminately explores all hues at various levels of saturation. The resulting invariance can be useful for downstream tasks where chromatic variations are indeed irrelevant (such as car body color in vehicle recognition), but will be detrimental for downstream applications where

the color information is known to be critical (such as flowers, birds, vegetables classification). On the other hand, completely removing color invariance risks producing a model with little generalization capability, unable to handle the common variations in illumination conditions due to various sources of indoor and outdoor lighting.

### 3.1 PHYSICS-BASED PLANCKIAN JITTER

Here we describe an alternative data augmentation procedure, called *Planckian jitter*, that exploits the physical description of a black-body radiator to re-illuminate the training images within a realistic distribution (Finlayson & Schaefer, 2001; Tominaga et al., 1999). The resulting augmentations are more realistic than those of the default color jitter (see Fig. 1). The resulting learned self-supervised feature representation is thus expected to be robust to illumination changes that can be commonly observed in real-world images, while simultaneously maintaining the ability to discriminate the image content based on color information.

Given an input RGB training image $I$, our physics-based Planckian jitter procedure applies a chromatic adaptation transform that simulates realistic variations in the illumination conditions:

1. We sample a new illuminant spectrum $\sigma_T(\lambda)$ from the distribution of a black body radiator.
2. We transform the sampled spectrum $\sigma_T(\lambda)$ into its sRGB representation $\rho_T \in \mathbb{R}^3$.
3. We create a jittered image $I'$ by reilluminating $I$ with the sampled illuminant $\rho_T$.
4. We introduce brightness and contrast variation, producing a Planckian-jittered image $I''$.

A radiating black body at temperature $T$ can be synthesized using Planck's Law (Andrews, 2010):

$$\sigma_T(\lambda) = \frac{2\pi hc^2}{\lambda^5 (e^{\frac{hc}{kT\lambda}} - 1)} \text{ W/m}^3, \tag{3}$$

where $c = 2.99792458 \times 10^8$ m/s is the speed of light, $h = 6.626176 \times 10^{-34}$ Js is Planck's constant, and $k = 1.380662 \times 10^{-23}$ J/K is Boltzmann's constant. For our experiments, we sampled $T$ in the interval between $3000K$ and $15000K$ which is known to result in a set of illuminants that can be encountered in real life (Tominaga et al., 1999). Then, we discretized wavelength $\lambda$ in 10nm steps ($\Delta\lambda$) in the interval between 400nm and 700nm. The resulting spectra are visualized in Figure 4 (left) in Appendix A.1.

The conversion from spectrum into sRGB is obtained through a series of intermediate steps (Wyszecki & Stiles, 1982):

1. We first map the spectrum into the corresponding XYZ stimuli, using the 1931 CIE standard observer color matching functions $c^{\{X,Y,Z\}}(\lambda)$, in order to bring the illuminant into a standard color space that represents a person with average eyesight.
2. We normalize this tristimulus by its $Y$ component, convert it into CIE 1976 L*a*b color space, and fix its L component to 50 in a 0-to-100 scale. This allows us to constrain the intensity of the represented illuminant in a controlled manner as a separate task.
3. We then convert the resulting values in sRGB, obtaining $\rho_T = \{R, G, B\}$. The resulting distribution of illuminants is visualized with the Angle-Retaining Chromaticity diagram (Buzzelli et al., 2020) in Figure 4 (right) in Appendix A.1.

All color space conversions assume a D65 reference white, which means that a neutral surface illuminated by average daylight conditions would appear achromatic. Once the new illuminant has been converted in sRGB, it is applied to the input image $I$ by resorting to a Von-Kries-like transform (von Kries, 1902), given by the following channel-wise scalar multiplication:

$$I'^{\{R,G,B\}} = I^{\{R,G,B\}} \cdot \{R, G, B\}/\{1, 1, 1\}, \tag{4}$$

where we assume the original scene illuminant to be white (1,1,1). Finally, brightness and contrast perturbations are introduced to simulate variations in the intensity of the scene illumination:

$$I'' = c_B \cdot c_C \cdot I' + (1 - c_C) \cdot \mu (c_B \cdot I'), \tag{5}$$

where $c_B = 0.8$ and $c_C = 0.8$ represent, respectively, brightness and contrast coefficients, and $\mu$ is a spatial average function.

## 3.2 LATENT SPACE COMBINATION

The self-supervised learning paradigm involves a pretraining phase that relies on data augmentation to produce a set of features with certain invariance properties. These features are then used as the representation for a second phase, where we learn a given supervised downstream task.

Default color jitter configurations generate features that are strongly invariant to color information, thus relying mainly on shape and texture for the downstream task. Constraining the color jitter consequently reduces the role of shape in favor of color in the decision process of the second phase. As stated by (Chen et al., 2020a), in fact, completely removing color jitter creates features that focus mainly on the color, and ignore the more difficult problem of learning a good shape representation. Our Planckian jitter is designed to operate in a middle ground, constraining the type of color transformation that should be given invariance to, and thus leaving space for shape information. Nonetheless, the relative role of color and non-color in the feature representation can be better tuned by creating a concatenation of features learned in different setups. For this reason, we propose a configuration where two pretraining phases are run in parallel: one with default color augmentation, which is expected to exploit mostly shape information, and one with Planckian jitter, which is expected to exploit color information. The two resulting features are then concatenated and used as the representation to learn downstream tasks (in our experiments each representation has 512 dimensions and the latent space combination 1024).

## 4 EXPERIMENTAL RESULTS

In this section we verify the superiority of the proposed color data augmentation method against the default color jitter and we ablate its components. We evaluate the impact on downstream classification tasks, and analyze the color sensitivity of the learned backbone networks. We have additional results on computational time of the proposed Planckian augmentation in Appendix A.4

## 4.1 TRAINING AND EVALUATION SETUP

We train a ResNet-18 (He et al., 2016) backbone encoder in the SimSiam framework and then evaluate it on a classification downstream task with a linear classifier. To analyze the impact of the augmentation operations on the learned features, we train the encoder with different data augmentations before the evaluation. See Figure 2 for a complete overview of the training procedure.

We perform the unsupervised training with two different image resolutions: $32 \times 32$ and $64 \times 64$ pixels[1]. To train the model at the lowest resolution, we adopted the CIFAR-100 dataset (Krizhevsky et al., 2009), which is composed of 50000 $32 \times 32$ images, divided into 100 classes. Since the training has been done with images of size $32 \times 32$, the ResNet18 architecture has been slightly modified: the first convolutional layer's kernel size has been modified from $7 \times 7$ to $3 \times 3$ and the first Max Pooling layer has been removed. Adopting the same modification to the architecture, we also trained a higher resolution version of the model, using the TINY-IMAGENET dataset (Le & Yang, 2015), which contains 100,000 images of 200 classes (500 for each class) at resolution $64 \times 64$ pixels. The linear classifier training at resolution $32 \times 32$ has been performed with two different datasets: the CIFAR-100 and the FLOWERS-102 datasets (Nilsback & Zisserman, 2008). The CIFAR-100 has been used in order to define a baseline for the classification task in generic conditions, since it has been used also in the original work on SimCLR. The FLOWERS-102 dataset, which contains 8189 photos of flowers, divided in 102 classes, has been selected to assess the quality of the features extracted in scenarios where color information covers an important role. The FLOWERS-102 dataset has been resized to dimension $32 \times 32$ to fit the pre-trained model input dimensions. For the training of the linear classifier at resolution $64 \times 64$, we used five different datasets: TINY-IMAGENET, FLOWERS-102 dataset, the VEGFRU dataset by Saihui Hou & Wang (2017), the CUB-200 dataset by Welinder et al. (2010) and the T1K+ dataset by Cusano et al. (2021). More details about these datasets are provided in Appendix A.2. In the case of CUB-200, each image has been cropped using the bounding boxes given in the annotations of the dataset. These five datasets have been resized at resolution $64 \times 64$ and used to train the linear classifier. For the T1K+, dataset we adopted the 266 class labeling to train the linear classifier and to test our models.

The model for supervised and unsupervised training has been written in Pytorch v1.7.0 and run on an NVIDIA Titan V 12 GB GPU. The SimSiam training has been performed using Stochastic Gradient

---

[1]Limited hardware resources hindered verification at standard ImageNet resolution ($224 \times 224$)

Table 1: Analysis of data augmentation. Self-supervised training is performed on CIFAR-100, whereas the learned features are evaluated at ($32 \times 32$) on CIFAR-100 and FLOWERS-102. Augmentation techniques include variations in hue and saturation (H&S), brightness and contrast (B&C), Planckian-based chromaticity (P), and random grayscale conversions (G). The accuracy refers to the results of the linear classifiers trained with features extracted from different backbones.

| | DATA AUGMENTATION | H&S | B&C | G | P | ACCURACY |
|---|---|---|---|---|---|---|
| CIFAR-100 | None | | | | | 41.93% |
| | Default Color Jitter | ✓ | ✓ | ✓ | | 59.93% |
| | Default Color Jitter w/o Random GrayScale | ✓ | ✓ | | | 41.96% |
| | Default Color Jitter (chromaticity only) | ✓ | | | | 32.46% |
| | Planckian Jitter (chromaticity only) | | | | ✓ | 36.10% |
| | Planckian Jitter (brightness/contrast only) | | ✓ | | | 31.78% |
| | Planckian Jitter | | ✓ | | ✓ | 47.31% |
| LSC | None + Default Color Jitter w/o Random GrayScale | ✓ | ✓ | | | 44.87% |
| | Default Color Jitter + Default Color Jitter w/o Random GrayScale | ✓ | ✓ | | | 62.27% |
| | Default Color Jitter + Planckian Jitter | | ✓ | | ✓ | 63.54% |
| FLOWERS-102 | None | | | | | 36.47% |
| | Default Color Jitter | ✓ | ✓ | ✓ | | 30.00% |
| | Default Color Jitter w/o Random GrayScale | ✓ | ✓ | | | 36.96% |
| | Default Color Jitter (chromaticity only) | ✓ | | | | 39.11% |
| | Planckian Jitter (chromaticity only) | | | | ✓ | 39.51% |
| | Planckian Jitter (brightness/contrast only) | | ✓ | | | 41.96% |
| | Planckian Jitter | | ✓ | | ✓ | 42.75% |
| LSC | None + Default Color Jitter w/o Random GrayScale | ✓ | ✓ | | | 43.33% |
| | Default Color Jitter + Default Color Jitter w/o Random GrayScale | ✓ | ✓ | | | 47.65% |
| | Default Color Jitter + Planckian Jitter | | ✓ | | ✓ | 51.66% |

Descent, with a starting learning rate $0.03$, cosine annealing learning rate scheduler, and mini-batch size 512 (like in original work Chen & He (2021)). The linear classifier training for CIFAR-100 is done with exactly the same settings, except a staring learning rate set to $0.1$ and a train process to 500 epoch. For other datasets we used Adam optimizer with initial learning rate set to $0.03$.

## 4.2 EXPERIMENTAL RESULTS - PLANCKIAN JITTER AUGMENTATION

### 4.2.1 ABLATION STUDY

To assess the impact of color data augmentations we define six different configurations:

- *None*: no color jittering operations (Random Color Jitter and Random Grayscale) are used.
- *Default Color Jitter:* the default configuration, as used in SimSiam and SimCLR, uses both Random Color Jitter and Random Grayscale operations.
- *Default Color Jitter w/o Random GrayScale*: same as *Default* without the Random Grayscale operation.
- *Planckian Jitter*: uses the complete proposed Planckian Jitter operation, operating on chromaticy and brightness and contrast aspects of the images. No Random Grayscale is applied.
- *Planckian Jitter (chromaticity only)*: applies Planckian jitter without modifying brightness and contrast values in the image.
- *Planckian Jitter (brightness/contrast only)*: this configuration only modifies images brightness and contrast values of the input images.

In all experiments these augmentations are combined with the other default augmentations, namely crop, horizontal and vertical flip and blur. Given these six configurations, six different models have been trained and have been fine tuned for image classification. For resolution $32 \times 32$ the model has been fine tuned with the datasets CIFAR-100 and FLOWERS-102. The results in terms of accuracy are reported in Table 1.

Looking at the results, we can identify basically two different trends for the two datasets. Considering CIFAR-100, as can be observed, removing the augmentations related to the colors makes the model less powerful, due to the loss of color invariance in the features extracted by the encoder, trained with contrastive learning approach. This behaviour is coherent to what was reported by Chen

et al. (2020a). As can be seen in Table 1, if color augmentations (i.e. brightness/contrast and Random GrayScale) are removed completely (*None* configuration), the accuracy drops of $18\%$. Furthermore, on the FLOWERS-102 the behaviour is the opposite with respect to CIFAR-100: removing the color augmentations helps the model to better classify images, obtaining an improvement of $6,93\%$ of accuracy. This behaviour confirms how the color invariance, learned in contrastive learning environment, negatively impacts downstream tasks where the color information has a more relevant role. For these scenarios a solution that takes into consideration the color information is preferred.

Considering the introduction of realistic color transformations, we can observe how this kind of augmentation operations positively impacts the contrastive training, producing models that achieve even better results with respect to the configuration without any kind of image color manipulation. As can be seen from Table 1, by simply reducing the jittering operation to influence brightness and contrast, leaving hue and saturation unchanged, yields another boost in accuracy of $5.49\%$. When we start modifying chromaticity values using a realistic approach, moving to *Planckian Jitter* augmentation, the final result is a boost of $6.28\%$ in accuracy with respect to the *None* configuration, for a total boost of $12.75\%$ in accuracy with respect to the *Default* configuration. Also on CIFAR-100 dataset an improvement of $5.38\%$ is given by the use of Planckian Jitter with respect to the configuration that does not use any color augmentation operation. Despite this improvement, in this scenario the contrastive training with the realistic augmentation does not give better results with respect to the *Default* configuration.

Given the results obtained using the data augmentations reported in Table 1, and given the considerations made in Section 3.2, we finally evaluate the impact of combining latent spaces from different backbones. The combination analyzed is obtained by concatenating the features extracted by the models trained with different configurations (e.g. *Default* with *Planckian Jitter*). We tested three different Latent Space Combinations (LSC): the None configuration with the Default Color Jitter w/o Random GrayScale, the Default configuration with the Default Color Jitter w/o Random GrayScale and the Default Color Jitter with the proposed Planckian Jitter. The results have been reported in Table 2. We can see the benefit coming from the use of a realistic augmentation instead of using the features coming from a non realistic ones. On both datasets, the *Latent space combination* with Default and Planckian Jitter configuration achieves the best results. On the original task, represented by the CIFAR-100 dataset, this combination achieves a total accuracy of $63.54\%$, a $3.61\%$ more with respect to the *Default* configuration and a $16.23\%$ more with respect to the *Planckian Jitter* configuration. Comparing to the LSC using the Default ColorJitter w/o Random Grayscale, the version with the proposed Planckian Jitter achieves a small improvement of $1.27\%$ in classification accuracy. On the downstream task, represented by the FLOWERS-102 dataset, the *Latent space combination* reaches an accuracy value of $51.66\%$: an improvement of $21.66\%$ and $8.91\%$ in accuracy respectively for the two original configurations. Comparing here with the LSC using the Default ColorJitter w/o Random Grayscale, the combination with Planckian Jitter achieve an higher result, with a bigger gap in terms of accuracy with respect to the CIFAR-100 scenario. Here the use of Planckian Jitter brings in an improvement of $4.01\%$, confirming the impact of using realistic augmentation in color related classification tasks. In both of the cases the LSC using the None configuration with the Default Color Jitter w/o Random GrayScale achieve the worst results out of the three LSC versions.

### 4.2.2 EVALUATION ON DOWNSTREAM TASKS

Given the results obtained from the ablation study, we performed the analysis of the proposed configurations on other downstream task, using the backbone trained on higher resolution images ($64 \times 64$ pixels). We reported here a subset of the previously tested configurations: *None*, *Default Color Jitter*, *Default Color Jitter without Random GrayScale*, *Planckian Jitter (brightness/contrast only)* and *Planckian Jitter*. We also reported the Latent space combination of the Default Color Jitter combined with the proposed Planckian Jitter.

Since the backbone for resolution $64 \times 64$ has been trained on TINY-IMAGENET dataset, we first fine-tuned and tested the behaviour of the different models on this dataset, then on the other ones representing the downstream tasks. As can be seen from the results in Table 3, the behaviour of the three models on the TINY-IMAGENET is the same we have seen on the CIFAR-100. This is expected since the tasks represented by these two datasets are similar in terms of relation between the color information and class discrimination. Looking at the other datsets, we can see that the introduction of the *Planckian Jitter* augmentation improves the accuracy on the test sets, and an even higher accuracy can be achieved by applying *Latent space combination*.

Table 2: Comparison training different contrastive learning models. Self-supervised training is performed on CIFAR-100, whereas the learned features are evaluated at $(32 \times 32)$ on CIFAR-100 and FLOWERS-102. We reported the best configurations obtained on SimSiam model, and retrained other two models, SimCLR and Barlow Twins, with those selected configurations.

| FRAMEWORK | DATA AUGMENTATION | CIFAR-100 | FLOWERS-102 |
|---|---|---|---|
| SimSiam | Default ColorJitter | 59.93% | 30.00% |
| | Planckian Jitter | 47.31% | 42.75% |
| | Latent space combination | 63.54% | 51.66% |
| SimCLR | Default ColorJitter | 56.99% | 35.29% |
| | Planckian Jitter | 47.75% | 45.00% |
| | Latent space combination | 61.07% | 55.78% |
| Barlow Twins | Default ColorJitter | 56.60% | 40.78% |
| | Planckian Jitter | 52.71% | 54.50% |
| | Latent space combination | 62.85% | 62.55% |

Table 3: Analysis on downstream tasks. Self-supervised training is performed on TINY-IMAGENET at $(64 \times 64)$. The comparison has been done considering multiple datasets as the downstream task.

| DATA AUGMENTATION | TINY-IMAGENET | FLOWERS-102 | CUB200 | VEGFRU | T1K+ |
|---|---|---|---|---|---|
| None | 27.06% | 37.65% | 18.76% | 24.07% | 35.82% |
| Default Color Jitter | 33.12% | 46.27% | 19.36% | 23.92% | 26.01% |
| Default Color Jitter w/o Random GrayScale | 31.62% | 40.39% | 21.90% | 27.39% | 32.50% |
| Planckian Jitter (brightness/contrast only) | 29.97% | 39.60% | 21.35% | 27.06% | 31.42% |
| Planckian Jitter | 30.95% | 52.35% | 25.12% | 28.94% | 32.51% |
| LSC (Default Color Jitter + Planckian Jitter) | 39.23% | 61.57% | 30.45% | 39.65% | 38.20% |

These experiments on different downstream tasks related to color and at higher resolution prove the effectiveness and the generalization capability of the proposed data augmentation and latent spaces combination. On the different tasks, the proposed data augmentation offers a reliable way to train a backbone feature extractor more effective with respect to the original set of augmentations. Moreover, with the Latent space combination we showed how the combination of the different feature spaces helps in the training of a more general model for image classification, since we obtain improvements not only on color related datasets, but also on the original general content datasets.

### 4.2.3 GENERALITY OF PLANCKIAN JITTER

To show that the method is generally applicable to self-supervised methods which exploit color jitter augmentation, we performed experiments on two other popular self-supervised models: the SimCLR and the Barlow Twins models. This comparison is reported in Table 2. As can be seen, for all of the three models tested, the application of the *Planckian Jitter* improves the accuracy on the downstream task. Independently from the model used, the *Default Color Jitter* configuration of data augmentation gives the worst results on the FLOWERS-102 dataset. This difference from the *Planckian Jitter* configuration is mainly related to the model sensitivity to color information: since the default configuration makes the different models invariant to color information, the classification on the downstream color-related task suffer, independently from the contrastive learning model adopted. An analysis on the color sensitivity of the different models is reported in section 4.3. The *Latent space combination (Default Color Jitter with Planckian Jitter)* configuration consistently achieves better results on both of the datasets. On FLOWERS-102 the final results improve considerably by over 20% for all three models.

### 4.3 COLOR SENSITIVITY ANALYSIS

In order to better understand the final models obtained by training the backbone architectures in the contrastive learning framework, we perform a robustness analysis on the FLOWERS-102 dataset with realistic illuminant variations and analyze the models sensitivity to color information.

Given the different backbones trained on CIFAR-100, we train the linear classifier on CIFAR-100 and FLOWERS-102 datasets, and test the models with different versions of these two datasets. Assuming as reference point the D65 illuminant, which for the purpose of this test is considered the default illuminant condition of each image in the datasets used for training and fine tuning, we

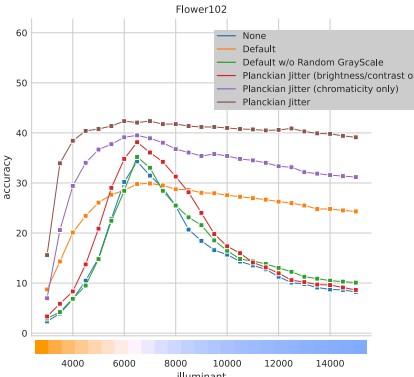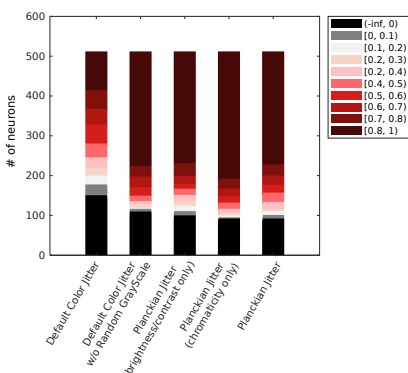

Figure 3: Color sensitivity analysis. On the left, the results of the robustness analysis: here are reported the accuracies at different illuminant achieved by the backbones trained with the different data augmentation configurations. On the right, the color sensitivity indexes computed for the different configurations used for training the backbone network.

change the illuminant of the two considered datasets. Using the 25 illuminants derived from the Planckian Jitter, we create different versions of each dataset, one for every illuminant. We are now interested in evaluating the robustness of the learned representations with respect to these illuminant changes. The results on Flowers-102 can be seen in Figure 3. Another plot containing also the accuracies on CIFAR-100 is provided in Figure 7 from Appendix A.3. The *Planckian Jitter* obtains stable performance from around 4000-14000K. On the other hand, the *Default Color Jitter* is more sensitive to the illumination color and the classification accuracy decreases when the scene illuminant is moving away from the white illuminant.

Finally, we conducted a color sensitivity analysis to inspect the impact of the color information for each neuron in the trained models. In order to perform the analysis we adopted the color selectivity index presented by Rafegas & Vanrell (2018). This index is defined as the property of a neuron to highly activate when color is present in the input images, and to not activate when color is absent. We computed the index for the last layer of the different backbones. More details on the color selectivity index can be found in Appendix A.3. The results are shown in Figure 3 and represent the amount of color sensitive neurons for each of the considered models. As can be seen, a major difference can be observed between the Default Color Jitter and the other proposed solutions: this difference in the sensitivity of the models is mainly related to the presence of the Random GrayScale augmentation in the *Default Color Jitter* configuration. This result confirms the hypothesis that models trained in this way are color invariant, a property that negatively affects the model in scenarios where color information has an important role, as demonstrated by our experiments. We can see also how the *Planckian Jitter (chromaticity only)* configuration is slightly more color sensitive with respect to the other configurations, due to the augmentation related only to realistic color variations in the scene.

## 5 CONCLUSION

Existing research on self-supervised learning mainly focuses on tasks where color is not a decisive component, and involuntarily exploits data augmentation procedures that negatively affect color-sensitive tasks, as we showed in this paper. We presented an alternative data augmentation technique based on the physical properties of light, called Planckian Jitter, and showed its positive effects on a wide variety of tasks where the intrinsic color of the objects (related to their reflectance) is crucial for discrimination, while the illumination source is not relevant, and as such the model should learn to be invariant to it. The general applicability of the proposed strategy for data augmentation is shown by experimenting with three different popular self-supervised models, demonstrating consistent improvements. We also proposed a solution that manages to exploit both color and shape information, by concatenating features learned with different modalities of self-supervision, achieving even superior performance. Finally, we presented an in-depth analysis on the sensitivity of trained neurons to color information, to demonstrate the difference in models trained with and without our Planckian Jitter. This produces a more general model for image classification, since we obtain improvements also on non-color-sensitive general datasets.

## REPRODUCIBILITY STATEMENT

The code of the Planckian Jitter data augmentation procedure, written in MATLAB and PyTorch 1.7.0, will be made available upon acceptance.

The training runs have been performed using Pytorch in combination with Pytorch Lightning Bolt framework, which provides an implementation of SimSiam methodology for backbone contrastive training. The model has been trained using CUDA deterministic, and random seed set to 1234.

All datasets used for the training and fine-tuning are publicly available. Only the CUB200 dataset has been pre-processed, by cropping each image using the given bounding box values, available alongside each image in the annotations files.

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

## A    APPENDIX A

### A.1    PLANCKIAN JITTER

Figure 4 illustrates the illuminants sampled from the distribution of a black body radiator, with correlated color temperature $T$ in the interval between $3000K$ and $15000K$. The resulting spectra are visualized on the left and in the middle, while the resulting distribution of illuminants is visualized with the Angle-Retaining Chromaticity diagram on the right.

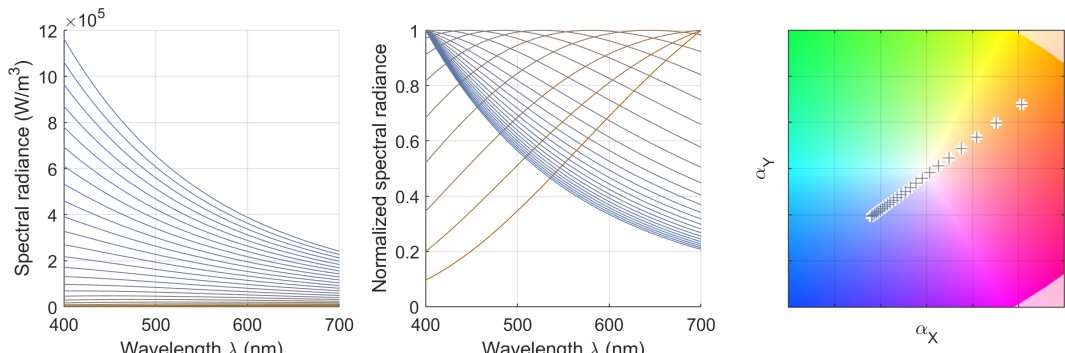

Figure 4: Spectral power distributions (left) and corresponding ARC chromaticities (right) of the sampled black body radiator, used to generate Planckian jittering.

Figure 5 shows a comparison between default color jitter (left) and Planckian jitter (right), replicating Figure 1 in xy chromaticity.

### A.2    DATASETS DETAILS

In section 4.2.2 we analyze the impact of our data augmentation, when using the features extracted by the backbone trained on TINY-IMAGENET dataset, on new datasets. The datsets used in the fine-tuning step are:

- FLOWERS-102 dataset by Nilsback & Zisserman (2008): dataset consisting of 102 flower categories, commonly occuring in the United Kingdom. Each class consists of between 40 and 258 images, for a total amount of 8.189 images.
- VEGFRU dataset by Saihui Hou & Wang (2017): this dataset consists of more than 160,000 images of vegetables and fruits, divided in 292 classes.
- CUB-200 dataset by Welinder et al. (2010): an image dataset made of 6033 photos of 200 bird species.
- T1K+ dataset by Cusano et al. (2021): a dataset of textures divided in 1129 classes, organized in 5 groups of 266 super classes. We adopted the 266 class labeling to fine tune and test our models.

Few example images for each of the color tasks datasets are reported in Figure 6.

### A.3    COLOR SELECTIVITY INDEX

Color selectivity is defined as the property of a neuron that activates highly when a specific color appears in the input image, and do not when the color is not present. The color selectivity index of a neuron is computed by estimating the ratio between its global activation with input color images and the global activation with corresponding grayscale images. By definition, the color selectivity index correspond to the complement to 1 of the ratio between the area under the activation curve given by the N-top grayscale images and the area under the activation curve given given by the N-top colored images.

$$\alpha(n^{L,i}) = \frac{\sum_{j=1}^{N} w'_{j,i,L}}{\sum_{j=1}^{N} w_{j,i,L}} \tag{6}$$

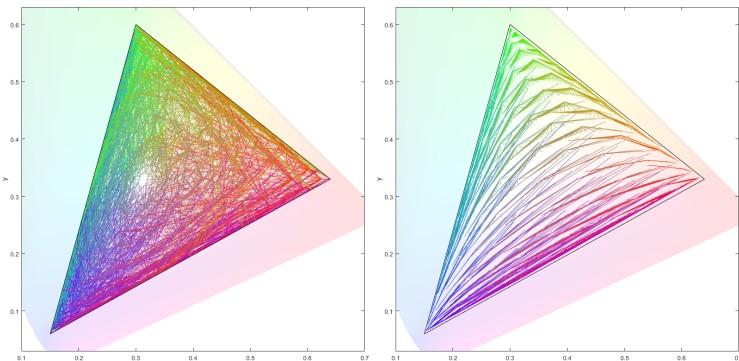

Figure 5: Default color jitter (left) and Planckian jitter (right) in xy chromaticity.

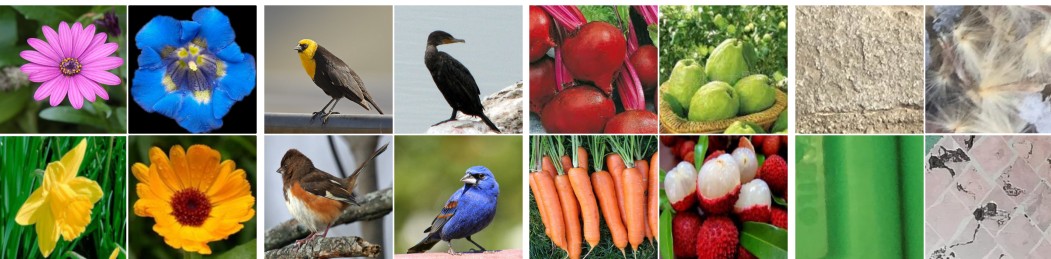

Figure 6: Example of images taken from the datasets used for the downstream classification task. From left to right the images are taken from FLOWERS-102 dataset, CUB-200 dataset, VEGFRU dataset and T1K+ dataset.

In the equation, $(w_{j,i,L})_{j=1:N}$ are the neuron activation values from the N-top colored images, while $(w'_{j,i,L})_{j=1:N}$ are the neuron activation values, by the same neurons, of the N-top grayscale images. We can distinguish between neurons that are color blind or neurons that highly relies on color information by looking at the alpha value obtained: an alpha value more than $0.25$ means that a specific neuron is high color selective, while an alpha value less than $0.1$ means that the neuron is basically color blind. These thresholds have been selected in relation to the work presented by Rafegas & Vanrell (2018). Using the definition of color selectivity index shown above, we collected alpha values for the neurons on the last layer of the encoders trained with different data augmentation configurations, in order to compare the models sensitivity to color and how it change in relation to the training procedure adopted.

To analyze the learned features robustness to different realistic illuminants, we tested the models with different versions of the CIFAR-100 and FLOWERS-102 datasets. We applied the *Planckian Jitter* on the two datasets, generating 25 different versions of each datasets, one for each illuminant sampled. Using these different versions of the datasets we then test the models for each illuminant and collect the classification accuracies. The results on both CIFAR-100 and FLOWERS-102 can be seen in Figure 7.

### A.4 EXECUTION TIME COMPARISON

Here we reported an execution time comparison performed to assess the usability of the proposed data augmentation, with respect to the already existent Color Jitter data augmentation algorithm.

In order to analyse such aspect of the algorithms, we executed the two algorithms, the Color Jitter image transform from PyTorch Torchvision package (respectively at versions v1.8.1 and v0.9.1) and the proposed Planckian Jitter at different image resolutions. For each resolution, we run the code 40 times and averaged the execution time. Results can be seen in Figure 8. The execution has been performed in CPU using a Intel i7-8700 processor. As can be seen, the proposed augmentation algorithm is faster with respect to the Color Jitter algorithm.

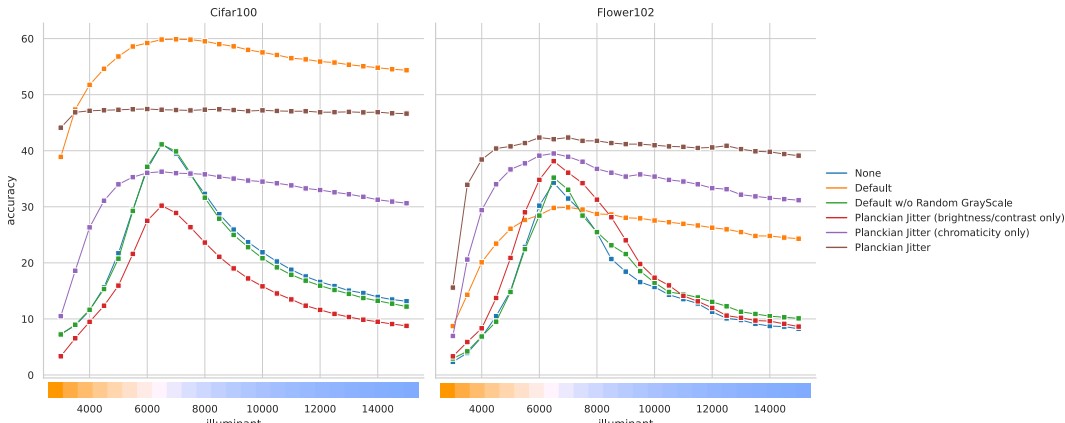

Figure 7: Illumimant robustness analysis. To assess the features invariance to realistic color changes in the images, for each method has been tested the classification accuracy with 25 different versions of the datsets. The images of the two datasets (CIFAR-100 on the left and FLOWERS-102 on the right) have been modified with the illumimants from temperature 3000 K to 15000 K, using the Planckian Jitter transform.

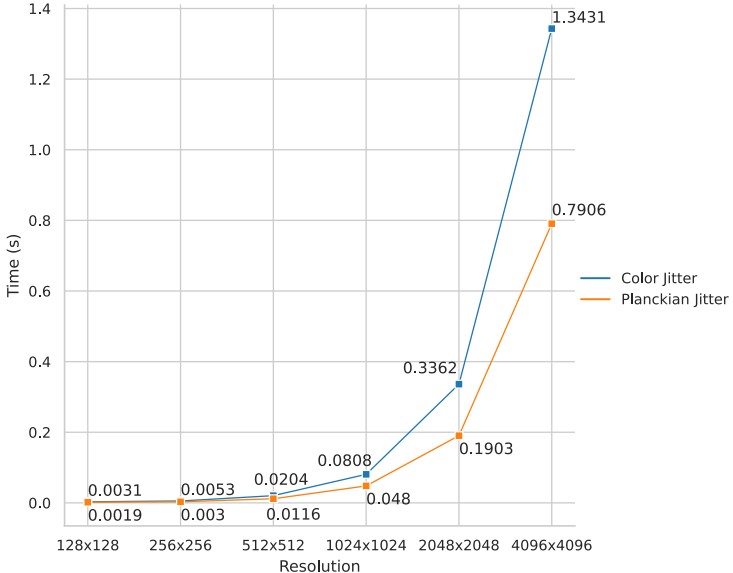

Figure 8: Execution time comparison between the proposed data augmentation algorithm implementation and the Color Jitter implementation in Pytorch Torchvision v0.9.1. For each resolution we executed both the algorithms 40 times.

