# OpenReview forum: "Planckian jitter: enhancing the color quality of self-supervised visual representations"
_ICLR.cc/2022/Conference — ICLR 2022 Submitted_

### Official Review · Reviewer_7zG7 · 2021-11-02

**Correctness:** 3
**Technical Novelty And Significance:** 3
**Empirical Novelty And Significance:** 2
**Recommendation:** 6
**Confidence:** 2

**Main Review:**

Strengths:
 - The proposed physics-based color augmentation is easy to understand and implement.
 - The performance seems to be surprisingly better than typical random color jittering.
 - This color augmentation can be applied to a wide variety of tasks.

Weaknesses:
 - Is the proposed physics-based color augmentation slower than typical color jittering. Many tasks need color augmentation performed on the fly during training. If the proposed color augmentation is much slower, the impact of this work will be marginal.

**Summary Of The Paper:**

This paper first examines that typical color jittering augmentation is harmful to feature representation learning. Then the authors proposed a physics-based color augmentation, called Planckian jitter to improve the performance. The proposed Planckian jitter performs better with the recent contrastive and self-supervised learning schemes.

**Summary Of The Review:**

This paper proposed a general color augmentation that performs better than the typical color jittering and demonstrates better performance in contrastive and self-supervised learning schemes. The only concern I have is the speed.

---

> ### Author Response · Authors · 2021-11-22
> **Response to Reviewer 7zG7**
>
> We would like to thank the reviewer for the comments.
>
> > The only concern I have is the speed.
>
> The proposed data augmentation is implemented in two stages:
> 1. A first offline stage, where we sample RGB illuminants from the Planckian Locus, by resorting to physics-based Equation 3. The final number of different illuminants can be varied parametrically, by defining the temperature interval and step (3000K-15000K with 500K step in our experiments).
> 2. An online stage, where we randomly select an illuminant from stage 1 for each image, and apply it by means of a simple per-channel scalar multiplication, following Equation 4.
> This makes the implementation particularly efficient, since the only operations done at training time are a random selection, and three scalar multiplications.
>
> To experimentally verify the efficacy of the proposed method, we tested the execution time of our Planckian Jitter augmentation, and the Color Jitter provided by PyTorch available in the torchvision module (configured as “Default Color Jitter” parameters in our manuscript).
>
> We report, in the following plot, the execution time by varying the image resolution, obtained by averaging 40 runs at each resolution. As can be seen our Planckian Jitter implementation is fast enough to be used as an online augmentation, and is even faster than the Color Jitter data augmentation provided by PyTorch.
> [Plot figure](https://i.imgur.com/ZyE73gS.png)
> We added an extra section in the Appendix (A.4) reporting those results.
>
> The default color jitter is slower than our solution, since their chromaticity variation is implemented by explicitly converting the RGB into HSV color space and back (for hue modification, see https://github.com/pytorch/vision/blob/8dcb5b810d85bd42edf73280db1ece38c487004c/torchvision/transforms/functional_pil.py#L83), and by computing a per-pixel linear combination between the original image and its grayscale version (for saturation modification, see https://github.com/pytorch/vision/blob/8dcb5b810d85bd42edf73280db1ece38c487004c/torchvision/transforms/functional_pil.py#L73 and https://pillow.readthedocs.io/en/stable/_modules/PIL/ImageEnhance.html#Color)

---

### Official Review · Reviewer_dWZ9 · 2021-11-03

**Correctness:** 2
**Technical Novelty And Significance:** 2
**Empirical Novelty And Significance:** 2
**Recommendation:** 5
**Confidence:** 4

**Main Review:**

**Pros:**

+ Well written and easy to follow manuscript
+The idea of the physics-based Planckian jitter which transforms colours along chromaticity lines is interesting
+ There does seem to be some support in the experiments for Planckian jitter improving results for colour-dependant domains (I.e. Table 1 when comparing "Default Color Jitter w/o Random GrayScale" and "Planckian Jitter")


**Cons:**

- Comparing the "Latent space combination" with other forms of augmentations seems very unfair since "Latent space combination" has double the capacity and requires double the training compute. To me "Latent space combination" seems more as an ensemble of models. To make it a fair comparison, it needs to be compared with other ensemble combinations (e.g. "Default Color Jitter w/o Random GrayScale"+"None" or "Default Color Jitter w/o Random GrayScale"+"Random Crop" seems like a reasonable baseline). Otherwise, I think "Latent space combination" should be removed altogether.
- Since cropping is one of the most important transformations for SSL, for all practical purposes there should be two versions of Table 1, one with and one without random cropping applied. I can imagine random cropping may impact the results.
- The main focus in current comparisons should be between "Default Color Jitter w/o Random GrayScale" and "Planckian Jitter", but this was left out entirely in Tables 2 and 3. Comparing "Default Color Jitter" (with Random GrayScale) against "Planckian Jitter" does not seem fair.
- Adding to above point, even comparing "Planckian Jitter" against "Default Color Jitter w/o Random GrayScale" isn't really enough. Since, as clearly indicated in Table 1, Planckian jitter mainly modifies brightness and contrast, there should be a direct comparison with an augmentation which changes brightness, contrast, brightness+contrast, and hue.
- There isn't any large scale datasets used in the experimentations, which raises the question of scalability and whether the Planckian augmentation would still be relevant with larger datasets.
- Planckian jitter is an augmentation technique which can also be evaluated on supervised learning tasks where image colours are important (and similarly, for transfer learning afterwards). Adding supervised experiments can add to the comprehensiveness of the experiments.
 - Section 4.2.1 mentions "As can be seen in Table 1, if color augmentations are removed completely (None configuration), the accuracy drops of 18%.". This seems incorrect when comparing "Default Color Jitter w/o Random GrayScale" to "None".
- The left plot of the colour sensitivity analysis (Fig 3.) seems unfair. The Planckian jitter models were specifically trained to ignore Planckian jitters, so one would naturally expect their results to be more robust with respect to Planckian jittering.


**Typos:**
- Section 3.1 paragraph 1 states "...default color jitter (see Fig. 3).", I believe this was intended to reference Fig 1.

**Questions for authors:**

1. It seems very odd to me that in table 1, "None" performs so similar to "Default Color Jitter w/o Random GrayScale". Do you have any thoughts on why this happens?
2. Were there any other augmentations used during training (cropping, rotations, etc)?

Edits:

Nov 3 -- typo fix

**Summary Of The Paper:**

**Overview.**
This paper proposes a novel type of image colour augmentation to be used during self-supervised learning (SSL).

**Background.**
In a typical SSL setting, similar samples are generated by randomly augmenting an image in a variety of different ways: random cropping, colour jittering, random rotations, etc. These similar images are then passed through a neural network, and the predicted features for similar samples are trained to be close to each other based on some similarity metric (ignoring the contrastive term description here as it is not directly related to the paper).

**Motivation.**
In this paper, the authors address the issue of using colour jittering as augmentation during SSL. Specifically, using colour jittering pushes the network towards invariance to image colours, while relying more on the shape and texture of objects when making predictions. The authors point out that despite the benefits of this for many general detection tasks, this will be a detrimental property when dealing with more colour-dependant tasks.

**Method.**
Thus, they propose to use Planckian jittering instead of colour jittering. Planckian jittering is a physics based augmentation method proposed by the authors (although the formulations for it come from existing literature) to re-illuminate the training images within a realistic distribution which leads to more realistic and constrained colour augmentations than colour jittering. The paper claims that Planckian jittering still helps improve network's dependence on shape and texture of objects (although less than colour jittering), while limiting the network's invariance to image colours.

**Experiments.**
6 SSL models were trained independently on CIFAR-100: 3 used different variants of the Planckian jittering, 2 used different variants of colour jittering (w/ and w/o Random GrayScale), and 1 used no augmentations. Linear classifiers for CIFAR-100 and FLOWERS-102 classification tasks were then trained on top of each of the SSL models' features, where FLOWERS-102 is the task that is claimed to be more heavily colour-dependant. Moreover, an extra linear classifier was trained on the concatenation of features from a Planckian jittering and a colour jittering model (called the "Latent space combination" model). Based on accuracy, "Latent space combination" outperforms all other models by a significant margin on both datasets. Planckian jitter seems to outperform other augmentations on FLOWERS-120 (Table 1), which supports the claim of the authors. On CIFAR-100, Planckian jitter performs slightly worse than colour jittering, which the authors attribute to the reduction of colour invariance in the features.

A very similar experiment was also done with different datasets (SSL training on tiny-imagenet, linear classifier trained on FLOWERS-102, CUB200, VEGFRU, T1K+), which also obtained similar results and conclusions. Moreover, in another similar experiment, the SSL models were trained with different SSL configurations (SimSiam, SimCLR, Barlow Twins) to indicate the generality of Planckian jitter for different types of SSL configurations. In another experiment, the robustness of the different models on augmented images using Planckian jittering was evaluated. Lastly, the colour sensitivity was analyzed to inspect the impact of colour information in neuron activations for each model.

**Summary Of The Review:**

This paper proposes an interesting idea and is very well written. The issues (with colour jittering) pointed out in this paper were previously known. Thus I view this paper as more geared towards practical purposes. However, the experiments are not comprehensive enough to strongly support the practical claims. Thus, my vote is for the paper to be rejected in its current form.

---

> ### Author Response · Authors · 2021-11-22
> **Response to Reviewer dWZ9 (Part 1)**
>
> We thank the reviewer for the feedback on our manuscript.
>
> > The issues (with colour jittering) pointed out in this paper were previously known.
>
> Self-supervised learning, based on networks that map two views generated by different data augmentations to the same latent-space point, has seen an enormous research effort in the last two years. Especially because these methods have been shown to close the gap with supervised learning without the need for large labelled datasets. This has led to hundreds of papers analyzing different aspects of the training pipeline. In our work, we show that for color-dependent downstream tasks replacing the default color jitter leads to a performance improvement of around 5% (Table 3). We also show that these representations are very complementary leading to gains around (10-15%). We are unaware of any prior work showing the color-crippling effect of the Default Color Jitter that is currently used in the vast majority of papers on this topic. Pointing out the consequences of using Default Color Jitter is one of the contributions of our paper; we genuinely think that this community is not aware of this effect. If the reviewer knows any papers that point out the detrimental effect of color jitter for self-supervised learning, we would be grateful if he could point us to them.
>
> > Comparing the "Latent space combination" with other forms of augmentations seems very unfair since "Latent space combination" has double the capacity and requires double the training compute. To me "Latent space combination" seems more as an ensemble of models. To make it a fair comparison, it needs to be compared with other ensemble combinations (e.g. "Default Color Jitter w/o Random GrayScale"+"None" or "Default Color Jitter w/o Random GrayScale"+"Random Crop" seems like a reasonable baseline). Otherwise, I think "Latent space combination" should be removed altogether.
>
> Doubling the capacity is not expected to yield large performance gains (See e.g. Fig. 4 of Barlow Twins paper [Zbontar et al.] where only Barlow Twins improves with 2% and the other two methods remain equal). We proposed the latent ‘space combination’ to show the complementary nature of the two representations: our method and the current method used in the vast majority of augmentation-based self-supervised learning methods.
> Our paper shows that the Standard Color Jitter is very harmful for the quality of the color representation. Based on this observation it makes sense to combine this Standard Color Jitter with an augmentation that allows for the learning of high-quality color representations. We agree that considering some other latent space combinations with those characteristics is interesting (but we would like to stress that these combinations are also a result from the observation made in our paper that the Default Color Jitter is harmful for the color quality; a fact that is unknown to the self-supervised learning community).
>
> We added two new configurations in the comparison of the models in Table1. As suggested we reported the results obtained by the latent space combination of the None configuration and the Default configuration without the use of Random Grayscale, and the combination of the Default configuration with the Default without Random Grayscale. This second configuration has been added in order to bring in the table the comparison between combining features  which mainly relies on structural information (Default configuration) and features with higher color dependency. The direct comparison with Default Color Jitter w/o Random GrayScale + Planckian Jitter comes naturally since here we are comparing the impact of the color dependent features once combined with structural related features. We added the comparison in Table 1 and commented on the new results in section 4.2.1.
>
> Concerning the Random Crop operation, we used it in all of the configurations presented in section 4.2.1. Since the focus of our work is related mainly to color features and color augmentations, we do not consider any configuration without random cropping or any kind of modifications in terms of spatial information augmentation.
> We added a comment in section 4.2.1 to clarify which augmentation are used in all of the configurations we presented in the same section.

---

> > ### Author Response · Authors · 2021-11-22
> > **Response to Reviewer dWZ9 (Part 2)**
> >
> > >Since cropping is one of the most important transformations for SSL, for all practical purposes there should be two versions of Table 1, one with and one without random cropping applied. I can imagine random cropping may impact the results.
> >
> > We use cropping in all our experiments. We have now made this more clear in the main manuscript (see Section 4.2.1, after the itemized list). We in fact use the default collection of data augmentations used by SimSiam model, which contains Random Cropping, and we only replace the Default Color Jitter. Since random cropping affects mainly spatial features and is not directly relevant for color, we do not perform any kind of variations on this kind of data augmentation operations in our experiments.
> >
> > We clarified the configuration of operations used in section 4.2.1.
> > Here an extract of the modified text (added text in bold):
> > "**In all experiments these augmentations are combined with the other default augmentations, namely crop, horizontal and vertical flip and blur.** Given these six configurations, six different models have been trained and have been fine tuned for image classification. For resolution 32×32 the model has been fine tuned with the datasets CIFAR-100 and FLOWERS-102. The results in terms of accuracy are reported in Table 1."
> >
> > >The main focus in current comparisons should be between "Default Color Jitter w/o Random GrayScale" and "Planckian Jitter", but this was left out entirely in Tables 2 and 3. Comparing "Default Color Jitter" (with Random GrayScale) against "Planckian Jitter" does not seem fair.
> >
> > We thank the reviewer for pointing out this lack in our comparison.
> > In order to enrich the comparison as proposed we have added in Table 3 other tree configurations: None, Default w/o Random Grayscale and Planckian Jitter (brightness/contrast only).
> > A comment to the added entries has been added in section 4.2.2.
> > "Given the results obtained from the ablation study, we performed the analysis of the proposed config-urations on other downstream task, using the backbone trained on higher resolution images (64×64pixels). **We reported here a subset of the previously tested configurations: None, Default Color Jitter, Default Color Jitter without Random GrayScale, Planckian Jitter (brightness/contrast only) and Planckian Jitter. We also reported the Latent space combination of the Default Color Jitter combined with the proposed Planckian Jitter.**"
> >
> > >Adding to above point, even comparing "Planckian Jitter" against "Default Color Jitter w/o Random GrayScale" isn't really enough. Since, as clearly indicated in Table 1, Planckian jitter mainly modifies brightness and contrast, there should be a direct comparison with an augmentation which changes brightness, contrast, brightness+contrast, and hue.
> >
> > In our experiments, we consider brightness+contrast variations together and ablate them for the Default Color Jitter as well as for our Planckian jitter. In our paper, we are especially interested in evaluating the effect of the unrealistic color jitter (hue-saturation variations).  We do not agree that Planckian jitter mainly modifies brightness and contrast only.  However, to make the comparison wider, we added in Table 1 the Default Color Jitter (chromaticity only) entry and to Table 3 three configurations which were also present in Table 1. Specifically, we added the None configuration, the Default Color Jitter w/o Random GrayScale and the Planckian Jitter which only modifies brightness and contrast values.
> >
> > >There isn't any large scale datasets used in the experimentations, which raises the question of scalability and whether the Planckian augmentation would still be relevant with larger datasets.
> >
> > We argue and experimentally verify that Default Color Jitter hurts the quality of the color features in the learned representations. We do not think that this detrimental effect is dependent on the dataset size. We show results on downstream tasks of varying size, including small (CUB, 6000 images) and large VegFru (with 160.000) images. On both these datasets we see similar improvements of around 5% for using Planckian Jitter.

---

> > > ### Author Response · Authors · 2021-11-22
> > > **Response to Reviewer dWZ9 (Part 3)**
> > >
> > > > Planckian jitter is an augmentation technique which can also be evaluated on supervised learning tasks where image colours are important (and similarly, for transfer learning afterwards). Adding supervised experiments can add to the comprehensiveness of the experiments.
> > >
> > > Self-supervised learning aims to map two different views (computed by taking two different data augmentations of the same image) to the same point in latent space. It was found that color jitter is a very important augmentation to obtain good results (see Fig. 5 and its discussion in ‘Simple Framework for Contrastive Learning of Visual Representations’). They reason that ‘Therefore, it is critical to compose cropping with color distortion in order to learn generalizable features.’  However, what was not well understood is that this led to a significant degradation of the quality of the learned color representations (the topic of our paper). In the next Section (Section 3.2) the same authors write ‘When training supervised models with the same set of augmentations, we observe that stronger color augmentation does not improve or even hurts their performance.’. We agree that it would be interesting to see if Planckian Jitter also has this detrimental effect in the supervised setting, but because we focus on self-supervised learning in this paper, we have not included those results and will consider them as future work.
> > >
> > > > Section 4.2.1 mentions "As can be seen in Table 1, if color augmentations are removed completely (None configuration), the accuracy drops of 18%.". This seems incorrect when comparing "Default Color Jitter w/o Random GrayScale" to "None".
> > >
> > > We consider Random GrayScale a color operation (since it changes the color content) and therefore compare Default Color Jitter with None (18%). We have clarified this in the text. The new sentence is ‘As can be seen in Table 1, if color augmentations (i.e. brightness/contrast and Random GrayScale) are removed completely (None configuration), the accuracy drops of 18%.’
> > >
> > > > The left plot of the colour sensitivity analysis (Fig 3.) seems unfair. The Planckian jitter models were specifically trained to ignore Planckian jitters, so one would naturally expect their results to be more robust with respect to Planckian jittering.
> > >
> > > We agree that Planckian jitter is trained to be robust to illuminant changes. Similarly Default Color Jitter is robust to hue/saturation changes. We argue that illuminant changes are more common in the world than hue/saturation changes, and that learning robustness with respect to hue-saturation changes can only be obtained at the cost of the quality of the learned color representation. Based on a large body of work (from color constancy theory) it has been verified that the illuminant changes we apply to the data model illuminant changes in the real world to a large extent. Therefore, Figure 3 shows that Default Color Jitter is not very robust to illuminant changes that can be encountered in the real-world, whereas the proposed method (as expected) is much more robust to these events.
> > > Also see the new illustration of illumination changes in the Supplementary section A.3
> > >
> > > >Section 3.1 paragraph 1 states "...default color jitter (see Fig. 3).", I believe this was intended to reference Fig 1.
> > >
> > > Thank you, we fixed the reference.
> > >
> > > > It seems very odd to me that in table 1, "None" performs so similar to "Default Color Jitter w/o Random GrayScale". Do you have any thoughts on why this happens?
> > >
> > > This behaviour can be related to the fact that color in cifar100 is not a strong discriminating feature, moreover the application of a transformation like the random color jittering is basically not helping in the self-supervised training, at least for these kind of datasets. The default version of the color jitter with the application of random grayscale makes the color invariance much stronger, as can be derived by looking at figure 3, where we showed that the number of neurons which are color sensitive is much smaller in comparison with other methods. This characteristic makes the features more suitable for this task. In fact by only applying the color jitter without random grayscale does not give an improvement in terms of accuracy with respect to the None version, where only flip, crop and blur are applied.
> > >
> > > > Were there any other augmentations used during training (cropping, rotations, etc)?
> > >
> > > We clarified which data augmentations are used alongside the different configurations reported in section 4.2.1.
> > > "**In all experiments these augmentations are combined with the other default augmentations, namely crop, horizontal and vertical flip and blur.** Given these six configurations, six different models have been trained and have been fine tuned for image classification. For resolution 32×32 the model has been fine tuned with the datasets CIFAR-100 and FLOWERS-102. The results in terms of accuracy are reported in Table 1."

---

> > > > ### Author Response · Authors · 2021-11-29
> > > > **Response to Reviewer dWZ9 (Part 4)**
> > > >
> > > > > 1. Comparing the "Latent space combination" with other forms of augmentations seems very unfair since "Latent space combination" has double the capacity and requires double the training compute. To me "Latent space combination" seems more as an ensemble of models. To make it a fair comparison, it needs to be compared with other ensemble combinations (e.g. "Default Color Jitter w/o Random GrayScale"+"None" or "Default Color Jitter w/o Random GrayScale"+"Random Crop" seems like a reasonable baseline). Otherwise, I think "Latent space combination" should be removed altogether.
> > > >
> > > > Regarding the first question, we've finished the experiments which were running and were not yet ready for the paper review deadline. We report here those results in order to provide a complete answer.
> > > >
> > > > |                           DATA AUGMENTATION                            | Tiny-ImageNet | Flowers-102 | Cub200 | VegFru |  T1K+  |
> > > > |------------------------------------------------------------------------|---------------|-------------|--------|--------|--------|
> > > > | None                                                                   | 27.06%        | 37.65%      | 18.76% | 24.07% | 35.82% |
> > > > | Default Color Jitter                                                   | 33.12%        | 46.27%      | 19.36% | 23.92% | 26.01% |
> > > > | Default Color Jitter w/o Random Gray Scale                             | 31.62%        | 40.39%      | 21.90% | 27.39% | 32.50% |
> > > > | Planckian Jitter (brightness/contrast only)                            | 29.97%        | 39.60%      | 21.35% | 27.06% | 31.42% |
> > > > | Planckian Jitter                                                       | 30.95%        | 52.35%      | 25.12% | 28.94% | 32.51% |
> > > > | LSC (Default Color Jitter + Default Color Jitter w/o Random GrayScale) | 39.02%        | 58.33%      | 26.82% | 36.43% | 37.20% |
> > > > | LSC (Default Color Jitter + Planckian Jitter)                          | 39.23%        | 61.57%      | 30.45% | 39.65% | 38.20% |
> > > >
> > > > We have added the latent spaces combination comparison with the version Default Color Jitter + Default Color Jitter w/o Random GrayScale, in order to compare the impact of the features obtainable by training the model with our proposed Planckian Jitter with respect to the default ones, in the version which is more focused on color (Default Color Jitter w/o Random GrayScale).

---

> > > > > ### Comment · Reviewer_dWZ9 · 2021-12-03
> > > > > **Response to rebuttal**
> > > > >
> > > > > Thank you for taking the time to write the detailed response. I think the paper is in a much better state now, so I'm increasing my score to "5: marginally below the acceptance threshold". I'm still not fully convinced by the experiments. I'm only replying to what I thought were the main points of the response.
> > > > >
> > > > > > Pointing out the consequences of using Default Color Jitter is one of the contributions of our paper; we genuinely think that this community is not aware of this effect.
> > > > >
> > > > > I disagree here -- it is known that positive samples in the contrastive loss build equivariance towards the augmentation used to generate those samples. By this token, if "Default Color Jitter" was used as augmentation, then the network would be more equivariant towards colour information in the images.
> > > > >
> > > > > > Doubling the capacity is not expected to yield large performance gains
> > > > >
> > > > > What you were doing is effectively a model ensemble, which would yield performance gains. The new added entries in Table 1 also support this: 'Default Color Jitter + Default Color Jitter w/o Random GrayScale' performs much more comparably now to 'Default Color Jitter + Planckian Jitter'
> > > > >
> > > > > > We have now made this more clear in the main manuscript (see Section 4.2.1, after the itemized list).
> > > > >
> > > > > Thank you for the positive change
> > > > >
> > > > > > ... we have added in Table 3 other tree configurations: None, Default w/o Random Grayscale and Planckian Jitter (brightness/contrast only).
> > > > >
> > > > > Thank you for the positive change
> > > > >
> > > > > > In our experiments, we consider brightness+contrast variations together and ablate them for the Default Color Jitter as well as for our Planckian jitter. In our paper, we are especially interested in evaluating the effect of the unrealistic color jitter (hue-saturation variations).
> > > > >
> > > > > Not fully convinced here -- I would really liked to have seen 'Default Color Jitter' entries in table 1 where either: 1- only the brightness was changes, 2- only the contrast was changed, 3- both the brightness and contrasts were changed
> > > > >
> > > > > > We do not think that this detrimental effect is dependent on the dataset size.
> > > > >
> > > > > Not convinced here -- this needs to be demonstrated via experimentations
> > > > >
> > > > > > Self-supervised learning aims to map two different views (computed by taking two different data augmentations of the same image) to the same point in latent space.
> > > > >
> > > > > Not convinced here -- Supervised learning also maps augmented images to the same point in space (since their label doesn't change through augmentation). So I'm not convinced that Planckian Jitter needs to be specific to SSL.

---

### Official Review · Reviewer_wAfa · 2021-11-04

**Correctness:** 4
**Technical Novelty And Significance:** 2
**Empirical Novelty And Significance:** 3
**Recommendation:** 6
**Confidence:** 5

**Main Review:**

Realistic data augmentation is important in self-supervised learning because this is the sort of data variability that one wants to ignore and be invariant to.
Overall, this is an example of physics-aware machine learning (or in simpler words, do things just right ;-) ). The proposed "Planck+VonKries" color augmentation transform is more than 100 years old, but it may be interesting for different audiences for different reasons: (a) people with color science background will not find it surprising, but they may be interested to see the quantitative gain that can be obtained from doing the classical right thing, and (b) the average computer scientist facing this problem for the first time will find a good visual motivation to avoid random augmentation (fig. 1) and a physically founded recipe to get a 10% improvement in classification.

Additionally, authors make an interesting observation: they argue that unrealistic (random) transforms may induce overinvariance (insensitivity) to certain aspects of the data, thus reducing the performance of the model or shifting the focus to other features of the data.
I think their results illustrate this general point: in their specific example, augmentation through unrealistic color jitter reduces the sensitivity to color (as confirmed by the number of color-sensitive neurons) and leads to extra consideration of spatial features.
A similar argument could be made for other augmentation strategies: if the database lacks natural variability, the discovered features are going to be unbalanced.

Overall I find this work as an interesting exercise in the context of data augmentation, although the technique is not novel and the general consequences are not extracted.

Other points to address:

* While the general change-of-focus effect is true, the authors should be more specific about when this is going to be a benefit. As it is presented now, it depends on the database or on the relevance of color for class discrimination.
My opinion is that the proposed augmentation (as any natural data augmentation) is going to be more positive in too artificial (too restricted) databases where illumination is fixed or it has too low variability. These are the cases that require the introduction of a natural variability (in this case of the illumination). This is consistent with the fact that wide databases such as CIFAR-100 as opposed to Flower102 get no improvement from the proposed jitter (see performance numbers or figure 6). Maybe this is because CIFAR-100 already includes the variability of natural illumination and hence explicit augmentation with the Planckian variability is not necessary.

Can the authors provide some extra evidence about the fact that neurons insensitive to color (when conventional color jitter is introduced) capture better the shape/texture information?

* Equation 4 for VonKries is confusing.
Von Kries-like adaptation consists of dividing each LMS (I may admit each RGB) value by the value corresponding to the white (or illuminant) in situation 1, and multiply by the value corresponding to the white (or illuminant) in situation 2 (see for instance Chapt.9 in Fairchild's Color Appearance Models, 2013).
This implies the multiplication of each tristimulus vector by clarify Eq. 4.

* The spectral radiances in Fig. 4 are confusing: using no normalization and a linear vertical axis implies that all the explored spectra seem "blueish" (with bigger contribution of short wavelengths). Decay in energy at low temperatures of the black body radiator make it difficult to see that between 3000 and 4000 K the contribution of long wavelengths is bigger ---> you have reddish illuminants too, but it is hard to see in that figure if you dont nromalize the energy or put a log scale in the vertical axis... Note that for instance the CIE A illuminant corresponds to 2856 K (almost the 3000K reported by the authors) and it is markedly reddish.

* Using the novel ARC chromatic diagram does not add anything special with regard to classical CIE xy diagram. Why not using the standard diagram?. Incidentally, I was surprised to see that the Planck locus is straight in ARC, but the variations in the colors in Fig. 1 (diagram at the right) go in curves... is this right?.

* Eqs. 1 and 2 seem to suggest that in Fig. 2 there is an additional Multilayer Perception in the lower branch of the training, isnt it?

**Summary Of The Paper:**

The authors propose a physically sensible data augmentation that takes into account the actual variation of day light illumination in self-supervised contexts.
They analyze the impact on the classification performance, the sensitivity of the performance to illumination changes, and the emergence of neurons sensitive or insensitive to color.


**Summary Of The Review:**

I find this work is an interesting exercise in the context of data augmentation (an illustration of how classical models from physics may benefit the learning), although the technique is not novel and the general consequences are not discussed in detail.

---

> ### Author Response · Authors · 2021-11-22
> **Response to Reviewer wAfa (Part 1)**
>
> We would like to thank the reviewer for the in-depth analysis of our manuscript.
>
>
> > Overall I find this work as an interesting exercise in the context of data augmentation, although the technique is not novel and the general consequences are not extracted.
>
> We have restructured and expanded the conclusions in order to better highlight the observations emerged from Section 4.2.3 (Generality of Planckian Jitter) and from the experiments conducted across multiple datasets. New text in bold.
>
> " Existing research on self-supervised learning mainly focuses on tasks where color is not a decisive component, and involuntarily exploits data augmentation procedures that negatively affect color-sensitive tasks, as we showed in this paper. We presented an alternative data augmentation technique based on the physical properties of light, called Planckian Jitter, and showed its positive effects on a wide variety of **tasks where the intrinsic color of the objects (related to their reflectance) is crucial for discrimination, while the illumination source is not relevant, and as such the model should learn to be invariant to it. The general applicability of the proposed strategy for data augmentation is shown by experimenting with three different popular self-supervised models, demonstrating consistent improvements.** We also proposed a solution that exploits both color and shape information, by concatenating features learned with different modalities of self-supervision. **This produces a more general model for image classification, since we obtain improvements also on non-color-sensitive general datasets.** "

---

> > ### Author Response · Authors · 2021-11-22
> > **Response to Reviewer wAfa (Part 2)**
> >
> > > While the general change-of-focus effect is true, the authors should be more specific about when this is going to be a benefit. As it is presented now, it depends on the database or on the relevance of color for class discrimination. My opinion is that the proposed augmentation (as any natural data augmentation) is going to be more positive in too artificial (too restricted) databases where illumination is fixed or it has too low variability. These are the cases that require the introduction of a natural variability (in this case of the illumination). This is consistent with the fact that wide databases such as CIFAR-100 as opposed to Flower102 get no improvement from the proposed jitter (see performance numbers or figure 6). Maybe this is because CIFAR-100 already includes the variability of natural illumination and hence explicit augmentation with the Planckian variability is not necessary.
> >
> > Thank you for the observations.
> > 1. Scope of the proposed augmentation:
> > We agree that the behavior of Planckian data augmentation depends on the downstream task, and our claim is that it works best for problems where the intrinsic color of the objects (related to their reflectance) is crucial for discrimination, while the illumination source is not relevant, and as such the model should learn to be invariant to it. This statement has been now included in the updated conclusions.
> > In our work we aim at producing color variations that are representative of real case scenarios, while respecting the sensitivity of intrinsic color information of certain tasks. The Flowers-102 dataset therefore appears to match both conditions for a representative test ground.
> > We concur that the proposed data augmentation would also be valuable in a scenario where the training dataset is too artificial/restricted, but we speculate that the difference in performance across the datasets used in our manuscript is not motivated by this (see rest of the response).
> > 2. Degree of variability of the datasets:
> > We show with samples from the Flowers-102 dataset that this is not necessarily a fixed-illumination (too artificial) dataset.
> > (Click link to see the images)
> > [Class 77](https://i.imgur.com/7uUbNPv.png), [Class73](https://i.imgur.com/gILH91g.png)
> > Below each image, we present a color patch extracted from the “supposedly white” petal regions, highlighting the variability in illuminant conditions. Please also note that, while blueish and yellowish lights are well represented by Planckian data augmentation, the last examples of green illuminants are extremely domain-specific, as they are introduced by surface inter-reflections from the leafy environment. As such, we consider these interesting for a characterization of the dataset heterogeneity, but we believe that they are too specific to be considered for a general-purpose solution.
> > Some degree of illuminant variation is, therefore, indeed already present.
> > Both the CIFAR-100 and Flowers-102, however, exhibit a degradation of classification accuracy when the model is trained with no data augmentation, and tested on a varying set of illuminants (See “None” curves in Figure 6 from the appendix). This suggests that, even if the original dataset contains some level of illuminant variability, this is not necessarily sufficient to cover consistent and strong illuminant variations (of the same nature), motivating the introduction of further augmentation.
> > 3. Difference in datasets performance:
> > We argue that the lower performance of Planckian data augmentation on the CIFAR-100 dataset is not necessarily determined by the dataset being more heterogeneous than Flowers-102. Instead, we claim it is given by the intrinsic nature of the task, where chromatic variations are irrelevant.
> > For example, the "car" class in CIFAR-100 has a wide variability of color bodies, and as such color information is not useful to correctly identify the class (the same can be expected for other classes in the dataset). Conversely, each flower species in Flowers-102 has one characteristic “reflectance” color, and any observed color variation is given by the illuminant.
> > This is consistent with what we show in Table 1:
> > In general, having random grayscale conversions (G) during contrastive pretraining makes color information unreliable (as it is not always available), so the resulting model is expected to be strongly color-invariant.
> > For CIFAR-100, removing this invariance drastically deteriorates the performance (from 59.93% to 41.96%), since the model learns color-sensitive features that lead to overfitting specific color instances seen during training.
> > For Flowers-102, removing this invariance improves the performance (from 30.00% to 36.96%), confirming the sensitivity of the task to color information.

---

> > > ### Author Response · Authors · 2021-11-22
> > > **Response to Reviewer wAfa (Part 3)**
> > >
> > > > Can the authors provide some extra evidence about the fact that neurons insensitive to color (when conventional color jitter is introduced) capture better the shape/texture information?
> > >
> > > We understand this is a reference to the following sentence from Section 3.2: “Default color jitter configurations generate features that are strongly invariant to color information, thus relying mainly on shape and texture for the downstream task”.
> > > Conceptually, a set of strong color variations is expected to reduce the features sensitivity to color (since such information is unstable and unreliable). The remaining pieces of information that can be used for class discrimination are thus related to shape and texture.
> > >
> > > Our claim is that, in this configuration, a larger portion of the feature space will be dedicated to non-color information. This is shown in Figure 3 (right), where the “Default color jitter” column has a large portion of color-insensitive neurons (black rectangle).
> > > We do not claim that the resulting features are also better at representing shape and texture information: we concur that this is plausible, since a larger feature space allows for a richer representation, but we believe that at this stage it is only a speculation, and it is not our intention to imply this.
> > >
> > > >Equation 4 for VonKries is confusing. Von Kries-like adaptation consists of dividing each LMS (I may admit each RGB) value by the value corresponding to the white (or illuminant) in situation 1, and multiply by the value corresponding to the white (or illuminant) in situation 2 (see for instance Chapt.9 in Fairchild's Color Appearance Models, 2013). This implies the multiplication of each tristimulus vector by clarify Eq. 4.
> > >
> > > As Situation 1 we assume a neutral illuminant (1,1,1) as we do not perform illuminant estimation during data augmentation.
> > > Situation 2 is the target of our data augmentation, so it corresponds to the chromaticity generated by the Planckian simulation.
> > > Based on the described adaptation, this corresponds to dividing by (1,1,1) and multiplying by the Planckian-generated R,G,B.
> > > We have now added the explicit division by the neutral vector in Equation 4.
> > >
> > > >The spectral radiances in Fig. 4 are confusing: using no normalization and a linear vertical axis implies that all the explored spectra seem "blueish" (with bigger contribution of short wavelengths). Decay in energy at low temperatures of the black body radiator make it difficult to see that between 3000 and 4000 K the contribution of long wavelengths is bigger ---> you have reddish illuminants too, but it is hard to see in that figure if you dont nromalize the energy or put a log scale in the vertical axis... Note that for instance the CIE A illuminant corresponds to 2856 K (almost the 3000K reported by the authors) and it is markedly reddish.
> > >
> > > We agree with the observation about the visualization of the radiance curves. We have considered both normalizing and changing the vertical scale to logarithmic, as the reviewer suggests, and eventually opted for normalization in the revised version.
> > >
> > > >Using the novel ARC chromatic diagram does not add anything special with regard to classical CIE xy diagram. Why not using the standard diagram?. Incidentally, I was surprised to see that the Planck locus is straight in ARC, but the variations in the colors in Fig. 1 (diagram at the right) go in curves... is this right?.
> > >
> > > 1. ARC allows for a direct visual observation of what eventually happens in the RGB color space, where the final color transformation takes place. On the other hand, xy chromaticity would represent the underlying physics in a more conventional and familiar way. We had originally considered the two alternatives, since both have their own advantages, and opted for the former within the main body of the manuscript. We have now included the corresponding plots in xy chromaticity as well in the appendix.
> > > 2. The Planckian locus appears straight (although not perfectly) as a combination of the choice of chromaticity, and the CIELab normalization step described in Section 3.1, which impacts the chromaticity as a byproduct of fixing the lightness.
> > > 3. Figure 1 is drawn by sampling N chromaticities, and connecting them to the corresponding variations given by our data augmentation. When the sample chromaticity is the neutral white, the resulting variations are consistent with Figure 4. But other samples produce stronger curvatures, which are also visually exaggerated by the overimposition of other samples with different slopes, which create the optical illusion of strongly curved variations.

---

> > > > ### Author Response · Authors · 2021-11-22
> > > > **Response to Reviewer wAfa (Part 4)**
> > > >
> > > > > Eqs. 1 and 2 seem to suggest that in Fig. 2 there is an additional Multilayer Perception in the lower branch of the training, isnt it?
> > > >
> > > > Equation 1 describes a loss function computed by alternating the inputs x1 and x2 between the two branches, but it is always only the first branch that uses a Multilayer Perceptron, producing either p1 or p2 depending on the alternation.
> > > > In this sense, what we draw in Figure 2 is only the first part of Equation 1.
> > > > We had adopted the original formalism from Chen et al. (2021) for both figure and equations, but we do agree that the resulting description is incomplete. We have now modified the manuscript by integrating this clarification, and we have further generalized Equation 2 to use different subscripts than Equation 1 and Figure 2.

---

### Decision · Program_Chairs · 2022-01-20

**Decision:**

Reject

**Comment:**

The reviewers were in general lukewarm about the paper, not convinced by why realistic augmentation mean more robust features in SSL, had concerns over the szie of the datasets (up to ~100k), and the success depends on the relevance of color for classification. The AC agrees with the reviewers. While the paper sounds interesting, there are many questions remain unanswered -- it's unclear that the rebuttal addressed the concerns shared by the reviewers.

In addition to the comments by the reviewers, the AC also feels that the overall design is adhoc and it's unclear that the proposed augmentation can generalize to larger, more practical problems.